# Speed of environmental change frames relative ecological risk in climate change and climate intervention scenarios

Daniel M. Hueholt [1] ✉, Elizabeth A. Barnes [1], James W. Hurrell[1] & Ariel L. Morrison[1]

Stratospheric aerosol injection is a potential method of climate intervention to reduce climate risk as decarbonization efforts continue. However, possible ecosystem impacts from the strategic design of hypothetical intervention scenarios are poorly understood. Two recent Earth system model simulations depict policy-relevant stratospheric aerosol injection scenarios with similar global temperature targets, but a 10-year delay in intervention deployment. Here we show this delay leads to distinct ecological risk profiles through climate speeds, which describe the rate of movement of thermal conditions. On a planetary scale, climate speeds in the simulation where the intervention maintains temperature are not statistically distinguishable from preindustrial conditions. In contrast, rapid temperature reduction following delayed deployment produces climate speeds over land beyond either a preindustrial baseline or no-intervention climate change with present policy. The area exposed to threshold climate speeds places different scenarios in context to their relative ecological risks. Our results support discussion of tradeoffs and timescales in future scenario design and decision-making.

The imprint of anthropogenic climate change is clear in ecosystems worldwide, with worsening impacts expected under all future emissions pathways[1–7]. High-impact risks such as these motivate the study of potential climate intervention methods to reduce climate impacts as efforts to decarbonize continue[8,9]. Stratospheric aerosol injection (SAI) is a hypothetical method to limit warming or cool the planet by adding reflective particles to the stratosphere[8]. Many different potential SAI deployment scenarios could complement emissions reductions. For example, SAI could be used to maintain global temperatures at or below some critical threshold or to rapidly reduce temperatures[8–12]. In contrast to carbon dioxide removal interventions, which operate on slower timescales[13], solar radiation management methods such as SAI currently represent the only known method to quickly alter global mean temperatures with near-future technology[8,9].

Species habituated to environmental niches must shift their range, adapt, or be extirpated as ambient conditions shift geographically in a changing climate[14,15]. The climate velocity of 2-meter temperature expresses the movement of thermal conditions and can be used to address the question: How fast, and in what direction, must an organism move over a period to stay in the same temperature conditions in which it started?[15,16]. Species have varying ability to shift their range in response to climate change; on average, marine organisms can move more quickly than terrestrial species, and trees have among the slowest responses of all forms of life[7,17,18]. Climate impacts to ecology emerge from many sources beyond temperature, including changes in precipitation[14], biogeochemistry[2,15], or interactions among species[19,20]. Species with very short life histories (e.g., bacteria) can adapt to a changing climate through evolution, while more complex organisms may be able to employ behavioral adjustments[1,14,21,22]. Populations unable to adapt or shift their range at sufficient rates may be at risk of extirpation–which often takes place abruptly following subsequent extreme events rather than as a slow, linear process accompanying the climatic change[1,23]. The climate velocity provides a general metric for perturbations to large-scale ecology by the

[1]Department of Atmospheric Science, Colorado State University, Fort Collins 80523 CO, USA. ✉e-mail: daniel.hueholt@colostate.edu

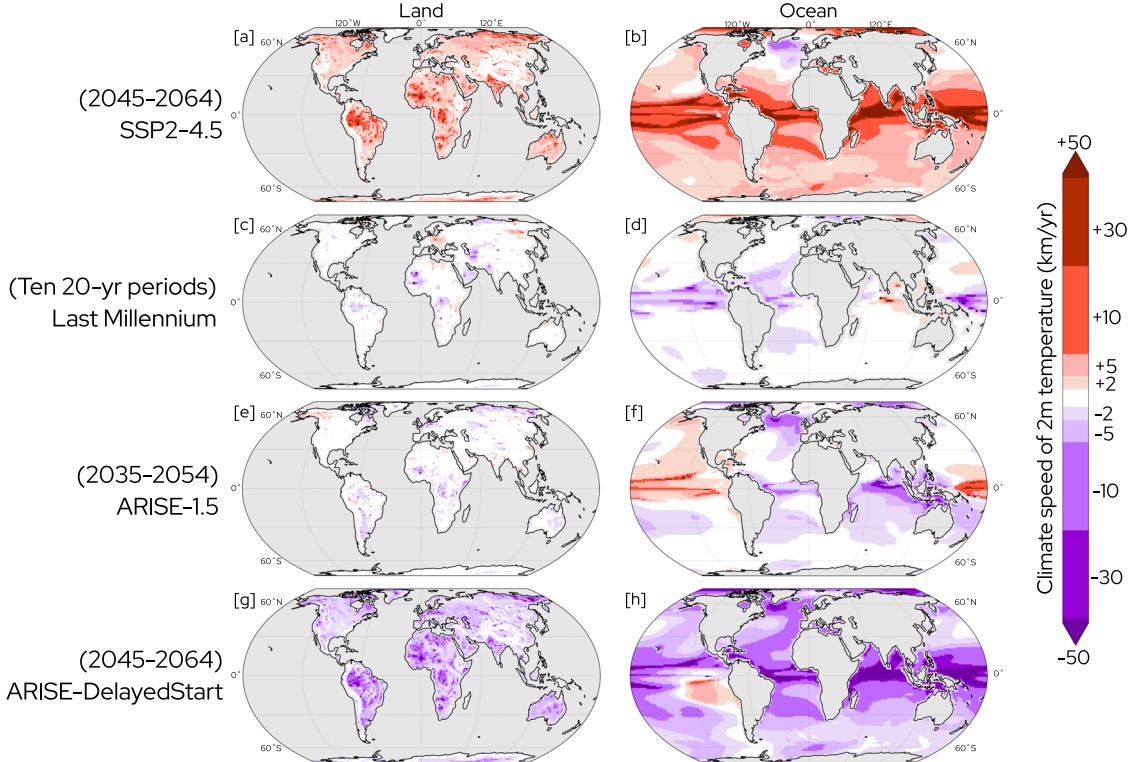

**Fig. 1 | 20-year climate speeds of 2-meter temperature on land and ocean.** 20-year climate speeds of 2-meter (2m) temperature on land (left column) and ocean (right column) in the ensemble mean for Shared Socioeconomic Pathway 2-4.5 (SSP2-4.5) (**a**, **b**), the mean of ten 20-year periods (to match ensemble size in Assessing Responses and Impacts of Solar climate intervention on the Earth system (ARISE) simulations, see "Methods" section) in the Last Millennium (**c**, **d**), and ensemble mean for ARISE-1.5 (**e**, **f**), and ARISE-DelayedStart (**g**, **h**). The sign indicates whether the change in temperature associated with the climate speed is positive or negative. Masked area is shown in gray (ocean for (**a**, **c**, **e**, **g**), land for (**b**, **d**, **f**, **h**)).

movement of thermal niches, rather than a tool to describe all types of impacts[15,16,19].

Future values of the scalar magnitude of climate velocity (which we refer to as the climate speed) under scenarios consistent with present policy exceed mean dispersal rates of known terrestrial ($\lesssim$2 km/yr[7,16]) and marine species ($\approx$7 km/yr[7]) and are expected to redistribute and endanger ecosystems globally[4,6,15]. The use of SAI has the potential to contribute an additional dimension of particularly rapid temperature change at the start or end of an intervention. Abrupt warming and cataclysmic climate speeds ("termination shock") are possible if an SAI intervention were to be terminated from masking a higher radiative equilibrium, or potentially dangerous cooling may occur at the start of an intervention intended to rapidly reduce global temperatures[18,24]. While the specific choices involved in generating a termination shock are clear[18,24,25], the strategic design decisions that could result in dangerous cooling rates are currently unknown.

We analyze climate speeds in the 20-year period following SAI deployment in simulations performed in the Community Earth System Model Version 2 with Whole Atmosphere Community Climate Model (CESM2[WACCM6]) under the Assessing Responses and Impacts of Solar climate intervention on the Earth system with stratospheric aerosol injection (ARISE) protocol[11,26,27]. ARISE simulations were constructed to allow outcomes to be directly connected to specific strategic design choices in each scenario. The ARISE-1.5 scenario portrays the deployment of SAI in 2035 to maintain the Paris Agreement global temperature target of 1.5 °C above preindustrial against moderate-mitigation climate change (Shared Socioeconomic Pathway [SSP] 2-4.5)[11]. ARISE-DelayedStart has a similar target of ≈1.37 °C but SAI deployment in 2045, yielding a rapid temperature reduction due to warming over the intervening decade[28]. A 10-year period represents a plausible delay that could come about from global governance and decision-making processes[12,29]. We compare these scenarios against baselines of preindustrial climate variability over the millennium prior to 1850 (Last Millennium, 850–1849) and no-SAI climate change consistent with present policy (SSP2-4.5)[26,30–33].

## Results

### Distinct responses linked to strategic choices

Global maps of climate speeds (Fig. 1) reveal highly distinct outcomes in the pattern of ecosystem risk in each of the four scenarios, reflecting their individual temperature trends over time (Fig. 2).

Substantial climate speeds forced by warming occur nearly globally under no-SAI SSP2-4.5 (Fig. 1a, b). The majority of land area (61%) is exposed to potentially dangerous climate speeds beyond 2 km/yr (Supplementary Fig. 1a). Very large climate speeds are projected to cause extreme ecosystem stress in tropical regions where spatial gradients are weak (Fig. 1a, b)[6,34]. For example, ensemble mean climate speeds averaged over the Amazon region (as defined by the IPCC Working Group 1 Fifth Assessment Report[35]) are 12 km/yr, suggesting that tropical terrestrial species would need to move poleward or up topography by 240 km in order to remain in their starting conditions over this 20-year period. Sharp topographic gradients buffer climate speeds and allow relict populations to shelter in microclimates, but these communities often have low connectivity[5,36] and persistent warming may render these niches inaccessible[5,16,36,37]. Poor connectivity can occur elsewhere due to causes including fragmentation by human land use such as urbanization[38,39], or natural barriers as in semi-enclosed marine basins like the Mediterranean Sea[40]. This fragmentation impedes the ability of many ecological communities to shift in response to climate changes and may increase population vulnerability[36,38,41].

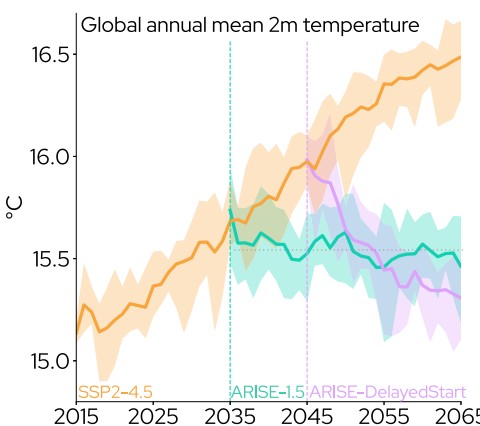

**Fig. 2 | Time series of global annual mean 2-meter temperature.** Time series of global annual mean 2-meter (2 m) temperature in the Shared Socioeconomic Pathway 2-4.5 (SSP2-4.5) and Assessing Responses and Impacts of Solar climate intervention on the Earth system (ARISE) 1.5 and DelayedStart simulations. Thick lines portray the ensemble mean; shading shows variability spanning the maximum to minimum ensemble member at each year. Vertical dashed lines denote the deployment of SAI in 2035 (ARISE-1.5) and 2045 (ARISE-DelayedStart), while the horizontal dotted line displays an approximate temperature threshold of 1.5 °C above preindustrial. Colors used to distinguish different simulations.

In the ocean, while depth gradients in temperature allow some species to escape climate change, non-thermal constraints prevent many from shifting vertically[15,34,42]. Climate speeds are large in the Arctic, where the warming rate is high due to Arctic amplification[43]. Transport barriers imposed by the edge of continents and the North Pole mark poleward limits on terrestrial and marine species[4,44]. Negative climate speeds occur in the North Atlantic warming hole (Fig. 1b) where a decreasing temperature trend is driven by the weakening Atlantic Meridional Overturning Circulation in these simulations[45,46]. Analogous to historical ecosystem responses to persistent internally-driven temperature anomalies[1,47], it is possible the North Atlantic may experience competing ecosystem responses between negative climate speeds associated with the warming hole and positive climate speeds elsewhere in the basin.

Climate speeds averaged over ten 20-year periods from the Last Millennium (Fig. 1c, d) are small, reflecting the relatively smaller magnitudes and slower evolution of climate forcings over this epoch. Volcanoes exert the largest external forcing on surface temperatures over the Last Millennium, but their influence is highly nonlinear and only persists for a few years[48,49]. As the climate velocity does not provide meaningful insight on these timescales[15], we omit periods within 5 years of a large volcanic eruption (defined as 10 teragrams of stratospheric sulfate injection[31]). Internally-driven climate variability[50,51], natural phenomena such as solar cycles[52], small volcanic eruptions[53], or anthropogenic land-use changes[54] can still cause nonzero regional-scale climate speeds (e.g., in Eastern Europe on Fig. 1c). Climate speeds are larger over the ocean (Fig. 1d) than land (Fig. 1c), reflecting the smaller spatial temperature gradients in marine environments[34]. Where temperature gradients are shallowest over the tropical oceans, even small perturbations to temperature can drive nonzero climate speeds[34]. The small magnitude of these climate speeds indicate internal variability and natural forcings over this period could lead to distributional shifts among species, but would not likely exceed their dispersal capabilities. A purely unforced simulation with boundary conditions fixed at 1850 (Supplementary Fig. 2) produces qualitatively similar results.

The 20-year climate speeds following deployment of ARISE-1.5 in the year 2035 (Fig. 1e, f) to maintain global mean temperature at 1.5 °C above preindustrial are relatively small compared to no-SAI SSP2-4.5

(Fig. 1a, b). These climate speeds reflect the nearly flat temperature trends implied by the use of SAI to maintain temperature (Fig. 2). Over land, climate speeds in ARISE-1.5 (Fig. 1e) are similar in magnitude to those in the Last Millennium simulation (Fig. 1c). Climate speeds over the ocean (Fig. 1f) are largely negative in sign. Since global temperatures are slightly above the 1.5 °C target when the intervention is deployed in 2035, ARISE-1.5 forces a small negative trend in temperature (Fig. 2). On regional scales, internal climate variability can overwhelm the forced response to the SAI intervention (e.g., Fig. 1f in the eastern Pacific)[55,56]. Negative climate speeds occur in the North Atlantic warming hole similar to no-SAI SSP2-4.5, as the weakening of the Atlantic Meridional Overturning Circulation is partially offset–but not halted–by the SAI intervention in ARISE-1.5[11,46,57]. Much more ocean area is exposed to climate speeds of 2 km/yr (48%) and 5 km/yr (17%) in ARISE-1.5 than the Last Millennium (23% and 8%, respectively). These values are within the observed mean dispersal rates of marine species (≈7 km/yr[3,7]), and little area is exposed to climate speeds that exceed these values in ARISE-1.5 (Supplementary Fig. 1b).

The SAI strategy in ARISE-DelayedStart produces large 20-year climate speeds (Fig. 1g, h) due to the negative temperature trend necessary to quickly reach the temperature target following deployment in 2045 (Fig. 2). A greater amount of land and ocean area is exposed to dangerous climate speeds in ARISE-DelayedStart (Fig. 1g, h) as opposed to no-SAI SSP2-4.5 (Fig. 1a, b; Supplementary Fig. 1). Two-thirds of land area (66%) is exposed to climate speeds beyond 2 km/yr; 13% of total land area (comparable to the size of South America) and more than a third of the world ocean (35%) are exposed to climate speeds greater than 10 km/yr in ARISE-DelayedStart (Supplementary Fig. 1). Five percent of the ocean is exposed to climate speeds beyond 50 km/yr (Fig. 1h, Supplementary Fig. 1b), which surpasses even the capability for extreme range shifts observed in many invasive species[58]. During the 20-year period following deployment, ARISE-DelayedStart depicts a forcing from climate speeds to global and regional ecosystems (Fig. 1g, h) that exceeds the corresponding time period in no-SAI SSP2-4.5 (Fig. 1a, b), and draws a striking contrast to the small values under ARISE-1.5 (Supplementary Fig. 3). This phenomenon of large climate speeds forced by rapid global temperature reduction could be viewed as a "deployment shock," similar to the termination shock previously identified if an intervention ceases at a high radiative equilibrium[18,24].

## Internal climate variability modulates conditions
Land and ocean median climate speeds in the Last Millennium simulation (Fig. 3a) illustrate the range of values experienced in 20-year periods in the preindustrial climate. The small magnitude of these climate speeds are within the range of dispersal rates for terrestrial and marine species[7,16]. When considering the full distribution of 20-year periods during the Last Millennium, climate speeds infrequently exceed mean dispersal rates of terrestrial species (≈2 km/yr) and never exceed those of marine species (≈7 km/yr) (Supplementary Fig. 9a). Median climate speeds over both the land and ocean from the ARISE-1.5 scenario where SAI is used to maintain global mean temperature fall within the distribution of Last Millennium variability (Fig. 3a). The global land and ocean median climate speeds in ARISE-1.5 are statistically indistinguishable from the Last Millennium simulation under a robustness test[56]. Climate speeds under no-SAI SSP2-4.5 robustly exceed both the Last Millennium and ARISE-1.5 over the land and ocean. In ARISE-DelayedStart, climate speeds surpass all other scenarios: the distribution is entirely separated from ARISE-1.5 or the Last Millennium, and robustly larger in magnitude than no-SAI SSP2-4.5 over land.

Contributions from internal variability are large on decadal to interdecadal timescales, such as the 20-year periods examined here, even in the presence of an external climate forcing such as an SAI intervention[50,55,56]. While analyzing the ensemble mean (Fig. 1) allows

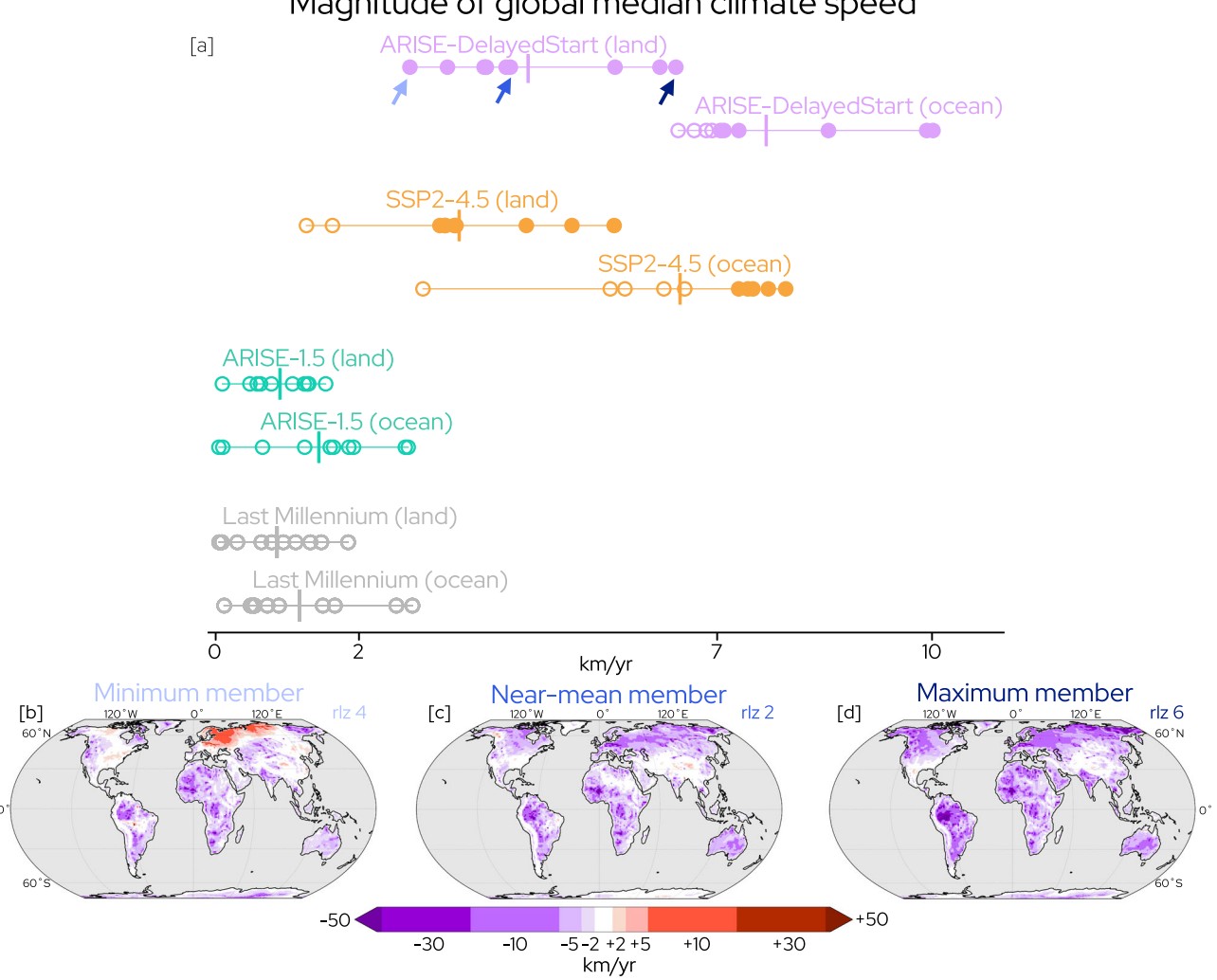

**Fig. 3 | Magnitude of global median climate speed and different realizations under internal variability.** Magnitude of global median climate speed of 2-meter temperature over land and ocean (**a**) in Shared Socioeconomic Pathway 2-4.5 (SSP2-4.5), Last Millennium, and Assessing Responses and Impacts of Solar climate intervention on the Earth system (ARISE) 1.5 and DelayedStart simulations. Maps of ensemble member with minimum (**b**), near-ensemble mean (**c**), and maximum (**d**) median climate speed over land in ARISE-DelayedStart. In [a], open circles denote climate speeds within the mean dispersal speed of terrestrial or ocean species, closed circles signify climate speeds exceeding mean dispersal speeds, and vertical bars show the ensemble mean. Arrows in (**a**) denote ensemble members (**b**–**d**). Climate speeds are calculated over 2035–2054 (ARISE-1.5), 2045–2064 (ARISE-DelayedStart and SSP2-4.5), and ten 20-year periods (Last Millennium). Colors in (**a**) distinguish different simulations. See Supplementary Figs. 4–7 for individual members in all simulations. Masked ocean area is shown in gray (**b**–**d**).

for investigation of the response to a climate forcing, each individual ensemble member (Fig. 3b, c, d) illustrates a plausible representation of the conditions that could be experienced under a single realization of internal variability. We describe the evolution of members across the ensemble of ARISE-DelayedStart to provide an example of the role of internal variability in the presence of a forced response to an SAI scenario of rapid temperature reduction. Climate speeds are large across the ensemble of ARISE-DelayedStart, exceeding dispersal rates in marine (≈7 km/yr) and terrestrial species (≲2 km/yr) in every ensemble member over land and six members over the ocean (Fig. 3a). Still, on regional scales in individual members, internal variability from sources such as the El Niño-Southern Oscillation or Pacific Decadal Variability can moderate trends or even flip their sign (Fig. 3b)[50]. Other members display a spatial pattern more similar to the ensemble mean (Fig. 3c). When average global trends from internal variability are in phase with the forced response, individual realizations can experience negative climate speeds of greater magnitude everywhere around the globe (Fig. 3d). In one realization of ARISE-DelayedStart, this amplification from internal variability

produces extreme median climate speeds over the global ocean exceeding 10 km/yr.

Previous analysis shows that the noise introduced by internal variability may impede detection of the surface climate response and lead to the perceived failure of an intervention[8,55,59]. The planetary-scale cooling in ARISE-DelayedStart is strong enough to entirely separate its distribution from no-SAI SSP2-4.5 when the sign of the trend is considered (Supplementary Fig. 9b); even in the member with the smallest global median climate speed, few regions see a sign opposite to the forced response (Fig. 3b). These results suggest that perceived failure at a regional or planetary scale would be much less likely under scenarios with rapid temperature reduction.

### Relative ecological risk from climate speeds

Climate speeds of 10 km/yr provide a threshold of extreme risk by exceeding the dispersal rates of both adaptable families (such as mammals) and terrestrial and marine species on average[7,16,18]. We plot the global area exposed to these climate speeds against the annual rate of global temperature change for a wide range of datasets to efficiently

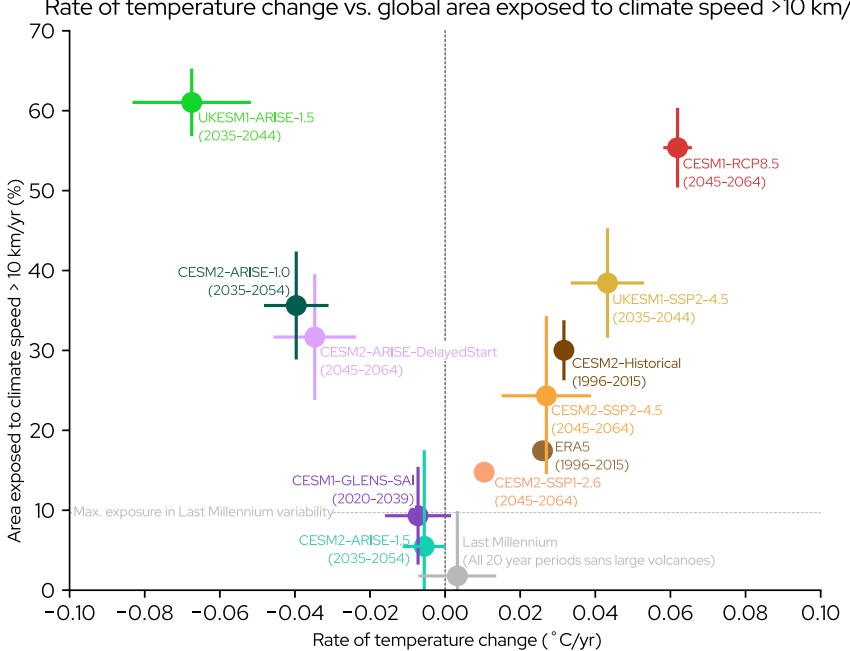

**Fig. 4 | Relative ecological risk given by rate of temperature change and global area exposed to climate speed beyond 10 km/yr.** 20-year rate of temperature change per year vs. percent of area exposed to a climate speed of 2-meter temperature with magnitude greater than 10 km/yr for various scenarios of climate change, climate intervention, and historical products. Dots denote the ensemble mean, and lines display the width of the ensemble variability. The colors of each dot help visually distinguish datasets from each other. Vertical dashed line shows 20-year change in temperature of 0 °C/yr. Horizontal dashed line represents the maximum 20-year area exposed to threshold climate speed in the Last Millennium variability (10%). See Table 1 for detailed descriptions of each dataset in figure, which are listed here from left to right: the United Kingdom Earth System Model 1 (UKESM1)-Assessing Responses and Impacts of Solar climate intervention on the Earth system (ARISE)-1.5, Community Earth System Model 2 (CESM2)-ARISE-1.0, CESM2-ARISE-DelayedStart, Community Earth System Model 1 (CESM1)-Geoengineering Large ENSemble (GLENS)-Stratospheric Aerosol Injection (SAI), CESM2-ARISE-1.5, Last Millennium, CESM2-Shared Socioeconomic Pathway 1-2.6 (SSP1-2.6), European Reanalysis 5 (ERA5), CESM2-SSP2-4.5, CESM2-Historical, UKESM1-SSP2-4.5, and CESM1-Representative Concentration Pathway 8.5 (RCP8.5).

summarize ecological risk (Fig. 4; see Table 1 for detailed data descriptions) and subsequently describe the implications for each product in the context of this figure.

The mean of all non-overlapping 20-year Last Millennium time periods is located at the origin, reflecting conditions that ecosystems experienced over the millennium before the Industrial Revolution began. The maximum area exposed (10% of global area) to the threshold climate speed over Last Millennium variability is denoted by the horizontal dotted line (Fig. 4). Greater distance from the origin beyond this dotted line denotes a relatively higher profile of ecological risk.

The European Reanalysis Version 5 (ERA5)[60] provides an observationally constrained global estimate of climate speeds during the recent past (1996–2015)[60,61]. Over this same period, a historical simulation (CESM2-Historical[27]) displays a larger area exposed to the threshold climate speed and a higher rate of temperature change than in ERA5. The physical reasons underlying the overly rapid warming rate in CESM2-Historical during this period are an ongoing area of research, and likely include errors in prescribed biomass burning emissions[62]. More generally, discrepancies between ERA5 and historical simulations may partly be due to structural differences between the single realization of real-world climate variability and the ensemble mean forced response[63]. Overall, both ERA5 and CESM2-Historical display climate speeds beyond Last Millennium conditions and within the range of mid-century SSP2-4.5, consistent with known historical and expected future ecosystem stress from warming[1,3,6,7,37,42].

No-SAI future scenarios expose substantial global area to large climate speeds from warming and cluster on the right half of Fig. 4. Scenarios with higher mitigation (SSP1-2.6), moderate mitigation (SSP2-4.5), and no mitigation (RCP8.5) all exceed the Last Millennium baseline, with increased emissions causing greater ecological risk. The

relatively higher risk portrayed in the SSP2-4.5 simulation in the United Kingdom Earth System Model version 1 (UKESM1-SSP2-4.5) (see "Methods" section for full description) as opposed to CESM2-SSP2-4.5 primarily stems from the more rapid warming rates due to the higher climate sensitivity in UKESM1[64]. The above results are in keeping with previous findings of widespread ecosystem stress under all future emissions pathways[1–7,37,42], with the most extreme risks from climate change under scenarios with no mitigation[34,65].

Scenarios where SAI is used for rapid temperature reduction cluster on the left side of the figure above the Last Millennium baseline and expose large amounts of global area to high climate speeds from rapid cooling. The individual scenarios (see Table 1 for details) each depict a unique potential design choice that could produce a deployment shock. One pathway would be through a delayed start with deployment after a temperature target has been surpassed, either through the choice to deliberately postpone deployment (ARISE-DelayedStart) or if the temperature threshold is breached early due to high climate sensitivity (UKESM1-ARISE-1.5[57]). Alternatively, the intervention could be deployed with the explicit goal of obtaining a low-temperature target below the starting global mean value (CESM2-ARISE-1.0[11,12], or the simulations of ref. 66 [not shown]). Regardless of the underlying strategic logic, the salient point is that every SAI scenario with rapid temperature reduction exposes more global area to the threshold climate speed than its corresponding no-SAI climate change reference scenario: CESM2-ARISE-1.0 and CESM2-ARISE-DelayedStart compared to CESM2-SSP2-4.5, and UKESM1-ARISE-1.5 compared to UKESM1-SSP2-4.5.

SAI scenarios where the intervention is used to maintain global mean temperature (CESM2-ARISE-1.5, CESM1-GLENS-SAI) remain near the origin, with ensemble means within the bounds of Last Millennium conditions. These SAI scenarios essentially eliminate the most extreme

**Table 1 | Table of all datasets used to calculate climate speeds**

| Name | Brief description | Time used | Resolution (lon × lat) |
|---|---|---|---|
| Last Millennium[26,30,31] | CESM2(WACCM6ma) simulation of the millennium prior to 1850 | Ten 20-year periods, and all non-overlapping 20-year periods | 2.5° × 1.89° |
| SSP2-4.5[11,32] | CESM2(WACCM6) simulation of future climate change with moderate mitigation and slow deployment of negative emissions technologies | 2045–2064 (10 ensemble members) | 1.25° × 0.9° |
| ARISE-1.5[11] | CESM2(WACCM6) simulation with SAI deployed in 2035 to maintain global mean temperature, pole-to-pole temperature gradient, and pole-to-equator temperature gradient at 2020–2039 mean against SSP2-4.5 forcing | 2035–2054 (10 ensemble members) | 1.25° × 0.9° |
| ARISE-DelayedStart[28] | CESM2(WACCM6) simulation with SAI deployed in 2045 to return global mean temperature, pole-to-pole temperature gradient, and pole-to-equator temperature gradient to 2020–2039 CESM1(WACCM5) mean against SSP2-4.5 forcing | 2045–2064 (10 ensemble members) | 1.25° × 0.9° |
| CESM1-GLENS[95] | CESM1(WACCM5) simulation with SAI deployed in 2020 to maintain global mean temperature, pole-to-pole temperature gradient, and pole-to-equator temperature gradient at 2010–2030 mean against RCP8.5 forcing | 2020–2039 (21 ensemble members) | 1.25° × 0.9° |
| ARISE-1.0[28] | CESM2(WACCM6) simulation with SAI deployed in 2035 to return global mean temperature, pole-to-pole temperature gradient, and pole-to-equator temperature gradient to 2000–2019 mean against SSP2-4.5 forcing | 2035–2054 (10 ensemble members) | 1.25° × 0.9° |
| UKESM1-ARISE-1.5[57] | UKESM1 simulation with SAI deployed in 2035 to return global mean temperature, pole-to-pole temperature gradient, and pole-to-equator temperature gradient to 2015–2034 mean against SSP2-4.5 forcing | 2035–2044 (5 ensemble members) | 1.875° × 1.25° |
| RCP8.5[95,102] | CESM1(WACCM5) simulation of future climate change with no mitigation | 2045–2064 (3 ensemble members) | 1.25° × 0.9° |
| UKESM1-SSP2-4.5[32,57] | UKESM1 simulation of future climate change with moderate mitigation and slow deployment of negative emissions technologies | 2035–2044 (5 ensemble members) | 1.875° × 1.25° |
| SSP1-2.6[32] | CESM2(WACCM6) simulation of future climate change with high mitigation and rapid deployment of negative emissions technologies | 2045–2064 (1 ensemble member) | 1.25° × 0.9° |
| CESM2-Historical[27] | CESM2(WACCM6) simulation of the climate state over the historical period | 1996–2015 (3 ensemble members) | 1.25° × 0.9° |
| ERA5[60] | Observationally constrained estimate of the historical Earth system | 1996–2015 (1 reanalysis) | 0.25° × 0.25°, remapped to 1.25° × 0.9° to match CESM simulations |
| Unforced[26,51] | CESM2(WACCM6) simulation of preindustrial conditions without external forcings | 10 randomly-selected 20-year periods | 1.25° × 0.9° |

risks to ecosystems from climate speeds occurring in the no-SAI climate change scenarios. However, a 10-year delay in deployment is the predominant difference between ARISE-1.5 and ARISE-DelayedStart. This short delay is sufficient to produce a highly distinct profile of extreme ecological risk.

## Discussion

This work demonstrates a key difference between scenarios where SAI is used to maintain global temperature, and those where SAI causes rapid temperature reduction. Scenarios that maintain global temperature greatly reduce risks from climate speeds, with global-scale parameters statistically indistinguishable from Last Millennium conditions. In contrast, rapid temperature reduction scenarios increase ecological risk ("deployment shock") relative to their corresponding no-SAI scenarios. The design of the ARISE scenarios allow these conclusions to be connected to specific potential decisions: a policy-relevant delay in deployment can turn a scenario that would otherwise greatly reduce ecological risk from climate speeds by maintaining temperature (ARISE-1.5) into one with a deployment shock that worsens this risk relative to no-SAI climate change (ARISE-DelayedStart).

Our results arise in policy-relevant scenarios designed for plausibility[11,12], as opposed to termination scenarios created to illustrate risks of SAI[24]. Deployment shock demonstrates a risk that intrinsically accompanies the ability to rapidly change temperature. This may restrict the ability to safely return to a temperature target after it has been surpassed. It is theoretically possible to design a strategy with sufficiently slow ramp-up of SAI to allow ecosystems to

respond to the forcing. However, SAI scenarios where global temperature is reduced are usually framed as an aggressive response option to relieve some severe impact of climate change[9,67], prevent tipping points[68], or to facilitate rapid detection by providing a large signal-to-noise ratio[12,59]. The strategic choice to slowly implement a low-temperature target may be in tension with these same goals.

Climate speeds are typically used for measuring ecosystem responses and risks in a warming climate[15,16,34], which raises the question of whether they are as meaningful for a cooling climate. Observed range shifts track temperature trends from internal climate variability regardless of their sign, strongly indicating both cooling and warming are ecologically relevant[1,47]. While relicts that temporarily survive warming through persistence or by sheltering in microclimates[36,37] would likely benefit from rapid cooling, numerous ecosystems that have transitioned to a new state under warming may be suddenly jeopardized. Paleoclimatic data indicates periods with rapid (interannual to multidecadal) large-scale cooling following a long-term warming trend coincide with planetary-scale changes to ecosystems[69,70]. These findings support the possibility that abrupt global cooling embedded in an antecedent warming trend could cause a large disturbance to ecosystems.

Insight from climate speeds can help inform future scenario design and decision-making. Designing scenarios to avoid deployment shock constrains both global temperature target and deployment year, which helps prevent a combinatorial explosion in scenario design[12]. We note two scenarios within these constraints that have not yet been simulated: delayed start maintenance with deployment dates past

2035 and higher temperature targets to avoid rapid temperature reduction, and slow starts where the intervention is implemented over sufficient time to moderate climate speeds. Decisions about global environmental policy involve complex tradeoffs of risk from many processes and phenomena[67]. As research concretely identifies sources of these tradeoffs in SAI scenarios, the relative prioritization of risks should be transparently documented during the design of a given scenario to help aid in analysis and effective decision-making.

## Methods

### Primary simulations

Our work draws on data from three simulations (SSP2-4.5, ARISE-1.5, and ARISE-DelayedStart) using the Community Earth System Model Version 2 with Whole Atmosphere Community Climate Model version 6 (CESM2[WACCM6])[26,27]. CESM2(WACCM6) is a fully interactive Earth system model with a high-fidelity depiction of the climate, including the stratospheric processes thought to be most relevant to SAI[26,27,71]. For all simulations here, CESM2(WACCM6) was run with 70 vertical levels (model top ≈140 km) and 1.25° longitude × 0.9° latitude horizontal resolution[11]. This spatial scale (Table 1) is considered adequate to analyze global ecosystem risk in the broader ecology literature[4,15,18].

The CESM2(WACCM6ma) Last Millennium dataset is a simulation of the 1000-year interval 850 through 1849, immediately preceding the Industrial Revolution which is defined to begin in 1850 by convention in the climate modeling community[26,30] Relatively abundant paleoclimate data allows for a well-constrained long-record depiction of this period including natural variability, realistic natural forcings including volcanoes and solar cycles, and anthropogenic land-use changes[31]. We use the Last Millennium to provide an ecologically relevant baseline of climate variability and change before anthropogenic climate change through greenhouse gas emissions and industrialization became large. CESM2(WACCM6ma) is a middle-atmosphere configuration of CESM2(WACCM6) and includes a simplified chemistry scheme to reduce computational complexity. The climate of CESM2(WACCM6ma) is very similar to CESM2(WACCM6) apart from the tropospheric chemistry[72].

The SSP2-4.5 simulations depict a no-SAI future with moderate mitigation of climate change and the slow deployment of negative emissions technologies[32]. Five ensemble members were created for the Coupled Model Intercomparison Project Phase 6[51]. An additional five ensemble members were created to augment the sample size for the ARISE project[11]. All 10 realizations are available from 2015–2069. SSP2-4.5 is consistent with present-day policy pledges by the global community, though it still results in warming beyond Paris Agreement targets in CESM2 and other climate models[33,73].

We use the ARISE-1.5 and ARISE-DelayedStart datasets to explore two policy-relevant SAI scenarios[11,28]. These simulations are often referred to as ARISE-SAI-1.5 and ARISE-SAI-1.37-DelayedStart. We use the names ARISE-1.5 and ARISE-DelayedStart for brevity, or CESM2-ARISE-1.5 and CESM2-ARISE-DelayedStart when necessary to distinguish from scenarios run in other models. SAI in ARISE-1.5 is deployed in 2035 to maintain global mean temperature at the 2020–2039 average in CESM2(WACCM6) (≈1.5 °C above the IPCC AR6 preindustrial value)[11,74]. In ARISE-DelayedStart, SAI is deployed 10 years later in 2045 with a similar global mean temperature target of the 2020–2039 average from CESM1(WACCM5) (≈1.37 °C above the IPCC AR6 preindustrial value) to depict the impacts of a policy-relevant delay in deployment[12]. ARISE-DelayedStart requires a larger stratospheric sulfate burden than ARISE-1.5 due both to the delayed start and the slightly lower temperature target[12,28].

Other design choices are constant between ARISE-1.5 and ARISE-DelayedStart: sulfur dioxide is injected at the same height (≈21 km), SSP2-4.5 greenhouse forcing is used in both, and each ensemble has ten members. Injections occur continuously from four locations (30° and 15° N/S, all at 180° E) with a proportional-integral feedback-control

algorithm to maintain the pole-to-pole and pole-to-equator temperature gradients alongside the global temperature target[11,75]. Controlling for these goals with off-equatorial injections is intended to reduce side effects by compensating for the planetary-scale spatial patterns of greenhouse warming: the increase in global mean temperature, hemispheric asymmetry, and polar amplification[76].

ARISE-1.5 and ARISE-DelayedStart are identical to SSP2-4.5 in every way except for the SAI intervention. Therefore, consistent differences between the simulations are likely due to the SAI strategies. The effect sizes of the SAI interventions in ARISE-1.5 and ARISE-DelayedStart relative to SSP2-4.5 are large enough that the global-scale results are clearly due to the SAI intervention (i.e., ensemble mean global temperature trend changing sign worldwide). Where useful, we additionally use the robustness test as a non-parametric method to identify where the forced response to the SAI intervention is large[56]. We refer to results as robust when they pass this test (corresponding to $p < 0.1$ under a binomial test).

The CESM2(WACCM6) Preindustrial control (Unforced) provides a single 500-year integration of the Earth system with perpetual 1850 greenhouse gas forcing[26]. This simulation illustrates the range of internal climate variability over an extended period of time without external forcings[51]. The small climate speeds in the Unforced simulation (Supplementary Fig. 2) raise confidence that the model is adequate for our analysis: while internal climate variability can produce pronounced ecosystem impacts in individual regions, planetary-scale risk to ecosystems would be implausible under unforced variability alone[1,47]. We use Unforced as a reference to perfectly-unforced conditions under internal variability alone, although land-use changes during and before 1850 imply it does not perfectly represent true equilibrium conditions[54].

### Climate velocity

The climate velocity of a geophysical quantity describes the movement of the isopleths of that variable in a changing climate[16]. Formally, the climate velocity is defined as the ratio of the temporal gradient of a variable $A$ ($\frac{dA}{dt}$, units time$^{-1}$) to the spatial gradient of that same variable ($\vec{\nabla} A$, units space$^{-1}$)[16]. The resultant variable ($\overrightarrow{C_A}$) has units of space per time–that is, a velocity (Equation (1))[16].

$$\frac{\frac{dA}{dt}}{\overrightarrow{\nabla} A} = \overrightarrow{C_A} \tag{1}$$

Climate velocity can be calculated for any variable but is most frequently applied to temperature[15]. Temperature exhibits a clear large-scale response to both climate change and SAI and has relatively well-understood spatiotemporal behavior in both observations and model output[15]. We use 2 m temperature rather than sea surface temperature over the ocean due to data availability limitations in ARISE-DelayedStart at the time of writing. On climatological spatiotemporal scales, 2 m temperature is similar to sea surface temperature and is often used for aquatic ecosystem analysis[77,78].

The climate velocity is a vector quantity, with both a magnitude and a direction. The scalar magnitude alone (climate speed) can be used separately from the vector quantity to quantify the high-level degree of disturbance to ecosystems[18,34,79,80]. This degree of disturbance is the quantity of interest for our research questions, and we use the climate speed exclusively in our analysis. We provide climate velocity vector maps for additional context (Supplementary Fig. 11), however, we caution readers that local analysis of these vectors would require a much finer-resolution dataset to better capture spatial gradients[15,16,40,80].

Following standard methods, we calculate the temporal gradient of temperature using linear regression and the spatial gradient of temperature using the 3 × 3 neighborhood slope algorithm[15,16,18,34]. In the accompanying Python code (fun_calc_var.py), we implement

the Sobel operator (mathematically equivalent to the 3 × 3 neighborhood slope algorithm) to obtain the spatial gradient of temperature. We calculate both our temporal gradient and spatial gradient directly from each dataset. At each point for each ensemble member, we divide the local 20-year temporal gradient (10-year for UKESM1 only, see below) by the spatial gradient and take the vector magnitude to obtain the climate speed. We impose a sign on the climate speed to denote whether it is associated with a warming or cooling trend. For all figures, we take the ensemble mean after the calculation of the climate speed. Climate speeds may be overestimated around complex topography in datasets with coarse spatial resolution[15]. Our results are robust to the choice of spatial resolution at the scale of all datasets used in this work (demonstrated for ERA5 in Supplementary Fig. 10).

By convention, climate velocities and climate speeds are assessed over time periods of 10 years or longer[15,16,18]. We calculate the climate speed over time periods chosen to be relevant to each scenario. For the scenarios with SAI in CESM2, this is the 20-year period immediately following deployment: 2035–2054 in ARISE-1.5, and 2045–2064 in ARISE-DelayedStart. 20-year timespans encompass the entire period when global mean temperature is decreasing in ARISE-DelayedStart. For no-SAI SSP2-4.5, we use the period 2045–2064 to compare to results from ARISE-DelayedStart. The time period 2035–2054 (corresponding to ARISE-1.5) is very similar in CESM2 simulations of SSP2-4.5, as the rate of change in global mean temperature does not alter substantially between 2035 and 2064. The time period spanning the interval when global mean temperature is decreasing in an SAI scenario is model-dependent and needs to be adjusted to correspond to the model that generated a given dataset. In UKESM1, a 10-year period fully encompasses the cooling after deployment due to its high aerosol sensitivity[57,81]. Thus, we calculate climate speeds over 10-year periods for output from UKESM1 as opposed to 20-year periods for all other products on Fig. 4. Using 20-year periods for UKESM1 would artificially reduce the climate speeds during its deployment shock by spreading the cooling out past the time horizon when global mean temperature has stabilized. In contrast, using the shorter 10-year periods for CESM2 would overlook the substantial cooling that continues past this horizon (Fig. 2).

The 10-member ensemble size of ARISE-1.5, ARISE-DelayedStart, and SSP2-4.5 enlarges the number of years available for analysis over each 20-year period to an effective size of 200 years[50]. Since the Last Millennium has only one ensemble member but a 1000-year simulation period, we choose ten 20-year time periods to avoid large volcanic eruptions (10 teragrams of stratospheric sulfate injection[31]): years 851–870, 871–890, 891–910, 911–930, 945–964, 971–990, 991–1010, 1011–1030, 1031–1050, 1051–1070. Avoiding such eruptions is necessary due to their large, but brief, impacts on global climate, which violate the linear trend assumptions underlying the definition of the climate velocity[15]. We additionally calculate the full distribution of non-overlapping 20-year climate speeds (again omitting large volcanic eruptions) in the Last Millennium for use in Fig. 4. Similarly, we choose ten 20-year time periods (years 5–24, 43–62, 95–114, 124–143, 164–183, 259–278, 280–299, 336–355, 379–398, 465–484) from the Unforced simulation to obtain a comparable sample size of 200 years, where relevant.

There is a wealth of ecological literature pertaining to the question of which climate speed values and periods of time correspond to ecosystem impacts on land and in the ocean. We cite only the most critical literature in the main body of the paper to remain within citation count restrictions, and provide references here for a fuller selection of this body of work for terrestrial ecosystems[7,16,34,82–85], marine ecosystems[3,7,34,82,84], and interannual to multidecadal range shifts[1,47,86–91].

### Additional data
We use a broad selection of data in Fig. 4 to discuss the relative risk between a variety of future scenarios of climate change and climate intervention and various depictions of the historical period. Table 1 enumerates all datasets used with a brief description of each.

## Data availability
The 2-meter temperature data from Earth system models and reanalysis used in this study (see Table 1 for compendium) have been deposited in the Open Science Framework database under accession code 10.17605/OSF.IO/Z37ES[92]. This archive includes all data used in our figures and analysis. Additionally, the complete raw datasets can be obtained at the following repositories and citations. ARISE-1.5 is available at the National Center for Atmospheric Research (NCAR) Climate Data Gateway under accession code 10.5065/9kcn-9y79[93]. CESM2-SSP2-4.5 is available at the NCAR Climate Data Gateway under accession code 10.26024/0cs0-ev98[94]. The CESM2 Last Millennium is available at the NCAR Climate Data Gateway under accession code 10.26024/5dgt-qf16: doi.org/10.26024/5dgt-qf16[30]. CESM1-GLENS-SAI and CESM1-RCP8.5 are available together at the NCAR Climate Data Gateway under accession code 10.5065/D6JH3JXX[95]. ARISE-DelayedStart and ARISE-1.0 are located on the NCAR Globally Accessible Data Environment file space while post-processing is conducted. The public permanent archive will be provided at the ARISE community page: cesm.ucar.edu/community-projects/arise-sai. ARISE-DelayedStart and ARISE-1.0 data used in our study is included at the Open Science Framework repository[92]. UKESM1-ARISE-1.5 is available at the Centre for Environmental Data Analysis under accession code 26b89d8d76bd40bfbaf9fedfa383e9cf: catalogue.ceda.ac.uk/uuid/26b89d8d76bd40bfbaf9fedfa383e9cf[96]. UKESM1-SSP2-4.5 is available at the World Data Center for Climate under accession code 10.26050/WDCC/AR6.C6SPMOU0[97]. CESM2-SSP1-2.6 is available at the World Data Center for Climate under accession code 10.22033/ESGF/CMIP6.10100[98]. CESM2-Historical is available at the World Data Center for Climate under accession code 10.22033/ESGF/CMIP6.10071[99]. The CESM2 Unforced (preindustrial control) is available at the World Data Center for Climate under accession code 10.22033/ESGF/CMIP6.10094[100]. ERA5 is available at the Copernicus Climate Data Store under accession code 10.22033/ESGF/CMIP6.10094[101].

## Code availability
Code used to process data and make all figures has been deposited at the Open Science Framework under accession code[92]. This code is licensed under the Open Software License 3.0.

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

## Acknowledgements

The authors thank Brian Dobbins, Alicia Karspeck, Jim Haywood, Matthew Henry, and Bette Otto-Bliesner for facilitating early access to data. Charlotte Connolly, Emily Gordon, and Alice Wells provided valuable comments as this work developed; Ezra Brody, Douglas MacMartin, and Daniele Visioni provided insight to the use of ARISE-DelayedStart. This study was supported by the National Science Foundation (NSF) Graduate Research Fellowship (Grant 006784) (D.M.H.) and the Defense Advanced Research Projects Agency (DARPA, Grant HR00112290071) (J.W.H., E.A.B., A.L.M.). The views, opinions, and/or findings expressed are those of the authors and should not be interpreted as representing the official views or policies of the Department of Defense or the U.S. government. D.M.H., J.W.H. and A.L.M. were also supported by the LAD climate group. ARISE-1.5, ARISE-DelayedStart, and ARISE-1.0 were produced and maintained by NCAR, and UKESM1-ARISE-1.5 and UKESM1-SSP2-4.5 by the UK Meteorological Office and University of Exeter. The ARISE project was supported by SilverLining through the Safe Climate Research Initiative. CESM2(WACCM6) Historical, Last Millennium, Unforced, and SSP1-2.6 were produced by NCAR. GLENS and RCP8.5 were supported by DARPA funding. The CESM project is supported primarily by NSF.

## Author contributions

D.M.H. conceived of the study, with input from A.L.M., E.A.B., and J.W.H. All data processing and code were developed by D.M.H. Scientific analysis was conducted by D.M.H. with insight and conceptual feedback from E.A.B., J.W.H., and A.L.M. Manuscript, figures, and supplementary information were written, and created by D.M.H. with editing by E.A.B., J.W.H., and A.L.M. D.M.H. revised the manuscript in response to reviewer feedback, with input from E.A.B., J.W.H., and A.L.M.

## Competing interests

The authors declare no competing interests.
