## [Peer Review File · Nature Communications]

Speed of environmental change frames relative ecological risk in climate change and climate intervention scenariosREVIEWER COMMENTS

Reviewer #1 (Remarks to the Author):

This is an interesting and important paper. The result that delaying implementation of SAI has great associated ecological risks is striking. The work is done very well, to the extent that I am able to assess. The question is of very broad general interest and is of value for ecologists, policy makers, environmental and conservation scientists, and to the general public.

Abstract:

Climate scientists call model runs with different parameters “experiments.” Ecologists, biologists more generally, and the public think of “experiments” as actual manipulations. This article will be likely to be read by a broader audience than just climate scientists. Given the fraught and contentious nature of the topic of climate intervention, it might be prudent to use a different word in the Abstract, and clarify how the term is being used in the body of the article, so that no one mistakes the work reported here as actually implementing SAI in the stratosphere.

The sentence that “climate speeds...are indistinguishable” depends on the scenario, but also depends on the resolution of the model. It may be true at some spatial scales for some scenarios (and models) for temperature, but it is definitely not true for precipitation and other aspects of climate, so while this statement is technically correct as stated, it is potentially misleading. One of the challenging things about both climate change itself and climate intervention for ecological systems are the creation of novel climate environments, where the combinations of climate components, yearly temporality, seasonality, and other aspects of climate are put together in novel ways. For example, apparently the SAI scenarios you are looking at might push global climates into a permanent and extreme La Nina state (from recent work presented at the 2023 AGU meeting). Who knows if that is correct, but stating that the earlier SAI intervention is indistinguishable from preindustrial climates really applies only to one scenario at one (pretty large) spatial scale and only for average annual temperature, and for ecological systems, that statement is misleading. Clearly the authors recognize the complexity of the changes that would occur with SAI and acknowledge it in the first couple of paragraphs of the introduction, but I think the encouraging result that annual average temperatures are restored if SAI is implemented sooner can be stated without being unintentionally implying something overly optimistic.

Because many readers are likely to summarily dismiss SAI, I have a few suggestions about things to add to highlight why SAI is really important to understand and consider. You might want to add a sentence or two to remind readers that planting large numbers of trees to sequester carbon is at best controversial, and at worst is not going to stave off climate change, while leading to many other (ecological and other) problems. I would also add a sentence to the effect that while scientists are working on carbon dioxide removal, and while it may one day prove to be an effective strategy for carbon reduction and climate cooling, we are very far from that as a viable solution now. That is, SAI has become a potential reality, so we had better take it seriously and understand its potential effects.

Text:

It is my understanding that delaying implementation also requires a larger SO₂ injection for the same cooling, and thus also has other additional risks to perturbing the climate system, e.g., ozone/UV changes (the more SO₂, the more these other perturbations). Not sure if that is what happens in the delayed scenario.

Line 74: Also, as for likely time needed before implementation, we need time for investigating the ecological and other consequences of SAI (including unintended consequences and risks), particularly with downscaling, and that means having time for funding agencies to start to recognize the imperative for funding research in this field. This has not yet happened at anything like scale (just a personal grumpy comment, not needed for the text or to be addressed by the authors).

Figure 1: Could these be complemented by additional maps showing vectors rather than colors? The colors are appropriately alarming for speed, but they of course do not indicate direction.

It would also be informative in the supplemental information to have maps with differences between the two ARISE implementations (2035 and delayed start).

Line 86-88: "...where spatial gradients are weak" is a really important point, and I get it, but it might be missed by general readers who have not been exposed to the climate velocity literature. Please spell this out more in an additional 1-2 sentences, with examples (e.g. a tropical tree in the Amazon Basin would have to move 15 km every year towards the equator to remain in the same temperature range). It is very surprising, and odd, that these extreme values are not the case for the boreal coniferous forests, which for large parts are really flat. Same, apparently, for the N.A. Great Plains and Eurasian steppe. The tundra is hit with warming, but not as much with cooling, except that little strip over the far northeastern Siberia. Why? I am puzzled and intrigued.

Also lines 86-88: Low connectivity, to say the least! To me, this is too understated and compresses the information so much that it is likely to be overlooked and below general readers' recognition. You might point out that global megacities, roads, and agriculture, not to mention the Gulf of Mexico and the Caribbean for NA, and the Mediterranean for Europe, are some of the tangible examples of why low connectivity is an issue.

In Figure 3, the colors did not have any obvious meaning, and the yellow was very hard to see.

Otherwise, it is a useful figure, although it would be less puzzling if it appeared after the explanation of the abbreviations in the text rather than before those explanations.

There are several ways for organisms and ecological systems (communities, forest types, etc.) to successfully respond to rapid climate change. Organisms can move, they can evolve, or they can employ plastic responses. It would be valuable to mention this in a short paragraph, because this is a paper focused on ecology, while having little actual ecology in it other than climate. Bacteria, viruses, and other organisms with very rapid life histories (including pathogens) can certainly evolve in response to climate change occurring over one or more decades. Insects can also probably evolve at those time scales, and some insects can move long distances quickly. Birds can move, at least some of them can, if the right habitats are somewhere they can reach and are available. Mammals, not so much. Some weed species can move quickly, but trees cannot (you might want to cite E.C. Pielou's classic work documenting tree movement following the North American glacial retreat after the last Ice Age—it took a long time, and was very different for different species). Many plants and some mammals have considerable plasticity, but perhaps not to sudden cold. I have no idea what happens to marine invertebrates, fish, phytoplankton, mammals, and others (there is a mention of that in the text, but not organism specific).

Because this paper is clearly aimed at ecologists and at a broader public, it would be valuable to give this broader context to those who have not been thinking about such scenarios.

I wish to sign my report.

Jessica Gurevitch, Purdue University

Reviewer #2 (Remarks to the Author):

This article discusses and expands upon the question of climate velocities in the context of climate intervention, comparing a set of datasets with different starting dates and cooling profiles, and contrasting the resulting regional climate velocities with those from climate change in the SSP2-4.5 scenario and in the recent past. I'll start by saying that the article is excellent: beautifully written, of high quality, and presenting novel and interesting analyses for one of the foremost issues in climate science in this decade. So I wholeheartedly endorse publication in Nature Communication. However, I think the analyses need to be complemented and expanded as I detail below before publication. I understand this will add work for the authors, and I am usually weary of asking more during review, but this paper has the potential to be a highly cited, essential part of the literature around SAI, and therefore I think the analyses need to be impeccable and include more than what has been included right now (especially for a NatComm study!).

One stylistic comment: Figure 1 is very hard to read, and should be greatly expanded. I can barely read the numbers on the colorbar even with zooming. Same for the supplementary figures with maps. Almost impossible to read...

For a reader not familiar with ARISE and SAI, imagining the difference between the various scenarios is hard. Yes, there's Figure 1 in the supplementary, but who reads that? My suggestion is to incorporate Fig 1 in the supplementary as the first panel of Fig. 1 in the main text (or on the left of it), so as to give even a cursory reader the proper context.

One minor comment: at line 262-263, the authors discuss the ability of return to lower temperature targets. I think the authors should cite this recent study (Visioni et al., 2023) <https://agupubs.onlinelibrary.wiley.com/doi/10.1029/2023EF003851> that shows results for exactly lower temperature targets. Considering these simulations also are performed with CESM2, they might even consider adding them to their Fig. 3, but it's not super necessary.

Now to the main issue:

I have an issue with using "preindustrial climate" in this context, which is highlighted by phrases such as "Preindustrial climate speeds (Fig. 1cd) are small, reflecting the absence of external forcings in this simulation.". This is true, but a reader should be disabused of the notion that this is what any "real" climate looks like, and then hence the climate velocities there are in any way what a species might expect from before 1850. PI is, indeed, a climate with no forcings at all. But regionally, in the last 2000 years, there have been other changes due to a variety of factors (some anthropics, some natural, such as one or more large volcanic eruptions) that have always to a point affected climate. Hence at the very

beginning I would at least be more explicit that the PI climate considered here is nothing but the best guess of CESM2 of what the internal variability of the climate looks like with no forcing. Another thing I think the authors should do is to integrate another dataset here, such as those provided by the pages 2k consortium (<https://www.nature.com/articles/sdata201788>) and repeat the same analyses they have performed over PI simulations also for temperature reconstructions of the last 2000 years. Only then will phrases like “Much more ocean area is exposed to climate speeds of 2 km/yr (48%) and 5 km/yr (17%) in ARISE-1.5 than Preindustrial (16% and 4%, respectively). These values are within the observed mean dispersal rates of marine species (≈ 7 km/yr [3, 7])” make sense, scientifically. This would greatly complement the analyses presented here and in Figure 3: their current dashed line in Figure 3 states that “Greater distance from the origin beyond this dotted line denotes a higher profile of ecological risk.” Which again, it is based on the assumption that the PI period is what species were led to (or better, evolved/adapted to) expect, and that suddenly in 1850 there was an abrupt shift. In order to make meaningful assumptions over things like that, one needs to also look at longer reconstructions (and not only things like ERA5 as done there, but something like the dataset I provided above).

Response to reviewers for Hueholt et al. "Climate speeds help frame relative ecological risk in future climate change and stratospheric aerosol injection scenarios" [Paper NCOMMS-23-51671-T]

General response

We are grateful to the reviewers for their thoughtful and constructive feedback, and were happy to see that both saw the relevance of our work to a broad audience. We have thoroughly revised the manuscript to address all comments. Changes in response to major comments include incorporating: a more appropriate preindustrial baseline in the CESM2 Last Millennium simulation (Reviewer 2), revising terminology for clarity to a broader audience (Reviewer 1), and adding further background information and analysis (Reviewers 1 & 2). Our detailed response to each reviewer comment is below. Changes to the manuscript can be viewed in the "tracked changes" version included with the revised submission. We additionally include passages from the text in individual comments as relevant.

Point-by-point response to Reviewer 1

This is an interesting and important paper. The result that delaying implementation of SAI has great associated ecological risks is striking. The work is done very well, to the extent that I am able to assess. The question is of very broad general interest and is of value for ecologists, policy makers, environmental and conservation scientists, and to the general public.

We thank the reviewer for these kind remarks, and are glad to see they find our work to be broadly meaningful.

Abstract:

Climate scientists call model runs with different parameters "experiments." Ecologists, biologists more generally, and the public think of "experiments" as actual manipulations. This article will be likely to be read by a broader audience than just climate scientists. Given the fraught and contentious nature of the topic of climate intervention, it might be prudent to use a different word in the Abstract, and clarify how the term is being used in the body of the article, so that no one mistakes the work reported here as actually implementing SAI in the stratosphere.

We have replaced "experiment" throughout the manuscript with terms such as "simulation" to emphasize these datasets are produced by numerical model runs with no real-world implementation. For example, the second sentence of the abstract now reads: "Two recent Earth system model simulations depict policy-relevant SAI scenarios with similar temperature targets (near 1.5 °C) but with deployment delayed by 10 years between each."

The sentence that "climate speeds...are indistinguishable" depends on the scenario, but also depends on the resolution of the model. It may be true at some spatial scales for some scenarios (and models) for temperature, but it is definitely not true for precipitation and other aspects of climate, so while this statement is technically correct as stated, it is potentially misleading. One of the challenging things about both climate change itself and climate intervention for ecological systems are the creation of novel climate environments, where the combinations of climate components, yearly temporality, seasonality, and other aspects of climate are put together in novel ways. For example, apparently the SAI scenarios you are looking at might push global climates into a permanent and extreme La Nina state (from recent work presented at the 2023 AGU meeting). Who knows if that is correct, but stating that the

earlier SAI intervention is indistinguishable from preindustrial climates really applies only to one scenario at one (pretty large) spatial scale and only for average annual temperature, and for ecological systems, that statement is misleading. Clearly the authors recognize the complexity of the changes that would occur with SAI and acknowledge it in the first couple of paragraphs of the introduction, but I think the encouraging result that annual average temperatures are restored if SAI is implemented sooner can be stated without being unintentionally implying something overly optimistic.

We agree that scenario- and model-dependence of results in SAI scenarios must be clear at all times. We have revised the abstract to clarify that the results in question are the planetary-scale climate speeds, which are inherited from the differing forced global temperature trends in the model simulations. The new text reads: "On a planetary scale, climate speeds in the simulation where global temperature is maintained with SAI are not statistically distinguishable from those experienced under preindustrial conditions." To avoid adding jargon to the abstract, we refrain from stating the full name of each simulation and instead describe the different forced temperature trends. This is why we state "the simulation where global temperature is maintained with SAI" and "scenario of rapid temperature reduction" rather than writing "ARISE-1.5" and "ARISE-DelayedStart."

For context, we note that the ENSO-locking results shown at AGU appear to be from simulations of marine cloud brightening (Xing et al. 2023). This is a different form of hypothetical climate intervention which is not discussed in our work and involves distinct processes from SAI.

Because many readers are likely to summarily dismiss SAI, I have a few suggestions about things to add to highlight why SAI is really important to understand and consider. You might want to add a sentence or two to remind readers that planting large numbers of trees to sequester carbon is at best controversial, and at worst is not going to stave off climate change, while leading to many other (ecological and other) problems. I would also add a sentence to the effect that while scientists are working on carbon dioxide removal, and while it may one day prove to be an effective strategy for carbon reduction and climate cooling, we are very far from that as a viable solution now. That is, SAI has become a potential reality, so we had better take it seriously and understand its potential effects.

We have added text in Section 1 contrasting the difference in timescale between the potentially rapid changes in temperature possible with SAI and the slow socio-physical processes of carbon removal. The passage now reads:

"For example, SAI could be used to maintain global temperatures at or below some critical threshold, or to rapidly reduce temperatures [8-12]. In contrast to carbon dioxide removal interventions, which operate on slower timescales [13], solar radiation management methods such as SAI currently represent the only known method to quickly alter global mean temperatures with near-future technology."

While we largely agree with the reviewer on the broader ecological, ethical, and governance issues pertaining to carbon removal, we have concerns that including a brief summary of this topic would risk coming across as superficial and may polarize readers. We feel a sufficiently nuanced and comprehensive discussion of these issues lies beyond the scope of this work.

Text:

It is my understanding that delaying implementation also requires a larger SO₂ injection for the same cooling, and thus also has other additional risks to perturbing the climate system, e.g., ozone/UV changes (the more SO₂, the more these other perturbations). Not sure if that is what happens in the delayed scenario.

Yes, a delayed start requires a larger sulfate burden than maintaining temperatures (MacMartin et al. 2022). In the specific case of ARISE-DelayedStart, this larger burden is due both to the later deployment and the slightly lower temperature target than ARISE-1.5. We have added a sentence to the Online Methods to state this explicitly. This text reads: "ARISE-DelayedStart requires a larger stratospheric sulfate burden than ARISE-1.5 due both to the delayed start and the slightly lower temperature target [11,12]."

The start time and aerosol burden interact within the Earth system in unintuitive ways. For example, delaying deployment results in a smaller impact on stratospheric ozone despite the larger sulfate burden—because the concentration of halogen species necessary for ozone destruction is projected to decrease between 2035 and 2045 (MacMartin et al. 2022). We refer to the papers documenting the ARISE-1.5 simulations (Richter et al. 2022) and ARISE-DelayedStart scenarios (Richter et al. 2022, MacMartin et al. 2022) with early simulations (MacMartin et al. 2022) for more information regarding the physical climate response.

Line 74: Also, as for likely time needed before implementation, we need time for investigating the ecological and other consequences of SAI (including unintended consequences and risks), particularly with downscaling, and that means having time for funding agencies to start to recognize the imperative for funding research in this field. This has not yet happened at anything like scale (just a personal grumpy comment, not needed for the text or to be addressed by the authors).

We thank the reviewer for this commentary!

Figure 1: Could these be complemented by additional maps showing vectors rather than colors? The colors are appropriately alarming for speed, but they of course do not indicate direction. *In the course of developing this study, we did create vector plots of the climate velocity. When we discussed this work in early forms, however, most audiences found the vector plots confusing and challenging to interpret. This was due to both the sheer quantity of vectors and the difficulty of visually comparing vector features between global-scale figures. We reproduce an example of a vector plot at the native grid resolution for the CESM2-SSP2-4.5 simulation here.*

Vector visualizations are clearer if the dataset is coarsened—however, the climate velocity directionality is only as accurate as the modeled spatial gradient. The nominal 1x1° resolution in the CESM2 SSP2-4.5 and ARISE simulations is already quite coarse with respect to ecosystems, or other important features such as orography or marine low cloud decks. Coarsening the data further dilutes these features. At 2.5x2.5° resolution the figures remain difficult to interpret, yet regions such as the Mediterranean Sea or Indonesia are visibly altered by the coarser grid. By 5x5°, the maps are more straightforward but phenomena as large as the North Atlantic warming hole have been compromised by the grid scale. We show examples of the CESM2-SSP2-4.5 simulation at 2.5x2.5° and 5x5° resolution below.

For completeness, we include vector maps below for SSP2-4.5, ARISE-DelayedStart, and ARISE-1.5 below at 2.5x2.5° resolution.

Vector analyses of climate velocity are frequent in regional-scale research (e.g., Burrows et al. 2011, 2014; Brito-Morales et al. 2018), and we imagine a future study using these methods in the context of SAI scenarios would yield interesting results. For the specific finding we wished to highlight here—the relative risk between scenarios arising from the differing global temperature trends—we felt the climate speeds stood better on their own. Thus, we continue to include only the climate speed figures within the manuscript.

It would also be informative in the supplemental information to have maps with differences between the two ARISE implementations (2035 and delayed start).

We have added a difference plot of ARISE-DelayedStart and ARISE-1.5 as Supplementary Fig. 3. This figure is now referenced in Section 2.1. The relevant text reads: "During the 20-year period following deployment, ARISE-DelayedStart depicts a forcing from climate speeds to global and regional ecosystems (Figure 1gh) that exceeds the corresponding time period in no-SAI SSP2-4.5 (Figure 1ab), and draws a striking contrast to the small values under ARISE-1.5 (Supplementary Fig. 3)."

Line 86-88: "...where spatial gradients are weak" is a really important point, and I get it, but it might be missed by general readers who have not been exposed to the climate velocity literature. Please spell this out more in an additional 1-2 sentences, with examples (e.g. a tropical tree in the Amazon Basin would have to move 15 km every year towards the equator to remain in the same temperature range).

We have added text to Section 2.1 explicitly spelling out the implications of the relationship between climate speed and the spatial gradient of temperature. The new passage reads:

“Very large climate speeds are projected to cause extreme ecosystem stress in tropical regions where spatial gradients are weak (Fig. 1ab) [6, 33]. For example, ensemble mean climate speeds averaged over the Amazon region (as defined by IPCC Working Group 1 Fifth Assessment Report [34]) are 12 km/yr, suggesting that tropical terrestrial species would need to move poleward or up topography by 240 km in order to remain in their starting conditions over this 20-year period.”

It is very surprising, and odd, that these extreme values are not the case for the boreal coniferous forests, which for large parts are really flat. Same, apparently, for the N.A. Great Plains and Eurasian steppe. The tundra is hit with warming, but not as much with cooling, except that little strip over the far northeastern Siberia. Why? I am puzzled and intrigued.

We agree these regional responses are puzzling and intriguing! Many of these features likely occur by chance due to stochastic climate variability opposing the regional forced temperature trend. It seems plausible that certain high-latitude features could be forced by the response to the pole-to-equator temperature gradient target in ARISE-DelayedStart—but this is purely speculation. Since our work is focused on global ecological impacts from the planetary-scale forcing, we do not attempt to explain regional results in depth. We also believe regional analysis would require much greater ecological and biological expertise than we possess as climate scientists, particularly given the limitations of climate velocity as a tool for projecting species-level impacts (e.g., Ettinger & HilleRisLambers 2013, Alabia et al. 2018). Further exploration of regional-scale effects of SAI on ecology, whether through climate speeds or other phenomena, is a clear avenue for future work. We hope other researchers will build on our results in this way.

Also lines 86-88: Low connectivity, to say the least! To me, this is too understated and compresses the information so much that it is likely to be overlooked and below general readers' recognition. You might point out that global megacities, roads, and agriculture, not to mention the Gulf of Mexico and the Caribbean for NA, and the Mediterranean for Europe, are some of the tangible examples of why low connectivity is an issue.

Yes! Our team had struggled to decide on what level of detail to provide here, and we appreciate the insight that this is a critical point. We have added text and references discussing connectivity in more depth. This new passage specifically includes the examples of both anthropogenic fragmentation and the Mediterranean Sea. The new text reads as follows:

“Sharp topographic gradients buffer climate speeds and allow relict populations to shelter in microclimates, but these communities often have low connectivity [5, 35] and persistent warming may render these niches inaccessible [5, 16, 35, 36]. Poor connectivity can occur elsewhere due to causes including fragmentation by human land use such as urbanization [37, 38], or natural barriers as in semi-enclosed marine basins like the Mediterranean Sea [39]. This fragmentation impedes the ability of many ecological communities to shift in response to climate changes and may increase population vulnerability [35, 37, 40].”

In Figure 3, the colors did not have any obvious meaning, and the yellow was very hard to see. Otherwise, it is a useful figure, although it would be less puzzling if it appeared after the explanation of the abbreviations in the text rather than before those explanations.

The yellow denoting CESM2-SSP1-2.6 on the figure (now Figure 4) has been changed to pink for better visibility. The colors are intended to help distinguish the dots representing different datasets, particularly where the bars representing ensemble uncertainty overlap (e.g., as for

CESM2-ARISE-1.0 and CESM2-ARISE-DelayedStart). We added text to the Figure 4 caption to clarify the colors distinguish differing simulations. This text reads: “Dots denote ensemble mean, and lines display the width of the ensemble variability. The colors of each dot help visually distinguish datasets from each other.” We evaluated multiple other approaches (e.g., all dots the same color and distinguished by the annotations, or using shades of black/gray rather than colors) and found these to be much more confusing.

We have revised the text introducing the figure in Section 2.3 to set expectations for the upcoming section, and provide the table reference for all datasets. The new passage reads:

“Climate speeds of 10 km/yr provide a threshold of extreme risk by exceeding the dispersal rates of both adaptable families (such as mammals) and terrestrial and marine species on average [7, 16, 18]. We plot the global area exposed to these climate speeds against the annual rate of global temperature change for a wide range of datasets to efficiently summarize ecological risk (Fig. 4; see Table 1 in Online Methods for detailed data descriptions), and subsequently describe the implications for each product in the context of this figure.”

There are several ways for organisms and ecological systems (communities, forest types, etc.) to successfully respond to rapid climate change. Organisms can move, they can evolve, or they can employ plastic responses. It would be valuable to mention this in a short paragraph, because this is a paper focused on ecology, while having little actual ecology in it other than climate. Bacteria, viruses, and other organisms with very rapid life histories (including pathogens) can certainly evolve in response to climate change occurring over one or more decades. Insects can also probably evolve at those time scales, and some insects can move long distances quickly. Birds can move, at least some of them can, if the right habitats are somewhere they can reach and are available. Mammals, not so much. Some weed species can move quickly, but trees cannot (you might want to cite E.C. Pielou’s classic work documenting tree movement following the North American glacial retreat after the last Ice Age—it took a long time, and was very different for different species). Many plants and some mammals have considerable plasticity, but perhaps not to sudden cold. I have no idea what happens to marine invertebrates, fish, phytoplankton, mammals, and others (there is a mention of that in the text, but not organism specific). Because this paper is clearly aimed at ecologists and at a broader public, it would be valuable to give this broader context to those who have not been thinking about such scenarios.

We thank the reviewer for raising these points. We have added text and citations to Section 1 (including the reference to Pielou 2008) providing more background information on how species respond to rapid climate change. The text reads:

Species have varying ability to shift range in response to climate change; on average, marine organisms can move more quickly than terrestrial species, and trees have among the slowest responses of all forms of life [7, 17, 18]. Climate impacts to ecology emerge from many sources beyond temperature, including changes in precipitation [14], biogeochemistry [2, 15], or interactions among species [19, 20]. Species with very short life histories (e.g., bacteria) can adapt to a changing climate through evolution, while more complex organisms may be able to employ behavioral adjustments [1, 14, 21, 22]. Populations unable to adapt or shift range at sufficient rates may be at risk of extirpation—which often takes place abruptly following subsequent extreme events rather than as a slow, linear process accompanying the climatic change [1, 23]. The climate velocity

provides a general metric for perturbations to large-scale ecology by the movement of thermal niches, rather than a tool to describe all types of impacts [15, 16, 19].

We acknowledge that this more granular information about biological responses to climate change falls outside our area of formal training as climate scientists. If elements of our added text are incorrect, we invite the reviewer to provide further context on these topics.

I wish to sign my report.

Jessica Gurevitch, Purdue University

We thank Dr. Gurevitch for her insightful review.

Point-by-point response to Reviewer 2

This article discusses and expands upon the question of climate velocities in the context of climate intervention, comparing a set of datasets with different starting dates and cooling profiles, and contrasting the resulting regional climate velocities with those from climate change in the SSP2-4.5 scenario and in the recent past. I'll start by saying that the article is excellent: beautifully written, of high quality, and presenting novel and interesting analyses for one of the foremost issues in climate science in this decade. So I wholeheartedly endorse publication in Nature Communication. However, I think the analyses need to be complemented and expanded as I detail below before publication. I understand this will add work for the authors, and I am usually weary of asking more during review, but this paper has the potential to be a highly cited, essential part of the literature around SAI, and therefore I think the analyses need to be impeccable and include more than what has been included right now (especially for a NatComm study!).

We thank the reviewer for their thoughtful review and constructive comments. In response, we have incorporated further data through the CESM2(WACCM6ma) Last Millennium simulation to provide a more appropriate baseline of preindustrial climate speeds, changed our terminology around preindustrial conditions, and added further analysis in response to both their comments and those of Reviewer 1.

One stylistic comment: Figure 1 is very hard to read, and should be greatly expanded. I can barely read the numbers on the colorbar even with zooming. Same for the supplementary figures with maps. Almost impossible to read...

We have redesigned Figure 1 to improve readability, and increased panel and text size in the supplementary figures as much as possible. With twenty individual maps for each scenario in Supplementary Figures 4-8, it is difficult to further increase the size without breaking these into an impractical number of individual figures.

For a reader not familiar with ARISE and SAI, imagining the difference between the various scenarios is hard. Yes, there's Figure 1 in the supplementary, but who reads that? My suggestion is to incorporate Fig 1 in the supplementary as the first panel of Fig. 1 in the main text (or on the left of it), so as to give even a cursory reader the proper context.

Supplementary Figure 1 has been added to the main text as Figure 2. We attempted to incorporate it into Figure 1, but this proved impractical while keeping each subfigure sufficiently large for readability (see response to previous comment).

One minor comment: at line 262-263, the authors discuss the ability of return to lower temperature targets. I think the authors should cite this recent study (Visioni et al., 2023) <https://agupubs.onlinelibrary.wiley.com/doi/10.1029/2023EF003851> that shows results for exactly lower temperature targets. Considering these simulations also are performed with CESM2, they might even consider adding them to their Fig. 3, but it's not super necessary.

We have added a reference to Visioni et al. 2023 to Section 2.3 as part of the discussion of design choices that may produce deployment shock. The text reads: "[...] the intervention could be deployed with the explicit goal of obtaining a low temperature target below the starting global mean value (CESM2-ARISE-1.0 [11, 12], or the simulations of [64] [not shown])."

We are reluctant to add more simulations in Figure 3 given the large number of datasets already present on the figure. Additionally, two of the three temperature targets in Visoni et al. 2023 are presently included in Fig. 3: their 1.5°C target matches ARISE-1.5, and the 1.0°C target ARISE-1.0.

Now to the main issue:

I have an issue with using “preindustrial climate” in this context, which is highlighted by phrases such as “Preindustrial climate speeds (Fig. 1cd) are small, reflecting the absence of external forcings in this simulation.”. This is true, but a reader should be disabused of the notion that this is what any “real” climate looks like, and then hence the climate velocities there are in any way what a species might expect from before 1850. PI is, indeed, a climate with no forcings at all. But regionally, in the last 2000 years, there have been other changes due to a variety of factors (some anthropics, some natural, such as one or more large volcanic eruptions) that have always to a point affected climate. Hence at the very beginning I would at least be more explicit that the PI climate considered here is nothing but the best guess of CESM2 of what the internal variability of the climate looks like with no forcing. Another thing I think the authors should do is to integrate another dataset here, such as those provided by the pages 2k consortium (<https://www.nature.com/articles/sdata201788>) and repeat the same analyses they have performed over PI simulations also for temperature reconstructions of the last 2000 years. Only then will phrases like “Much more ocean area is exposed to climate speeds of 2 km/yr (48%) and 5 km/yr (17%) in ARISE-1.5 than Preindustrial (16% and 4%, respectively). These values are within the observed mean dispersal rates of marine species (≈ 7 km/yr [3, 7])” make sense, scientifically. This would greatly complement the analyses presented here and in Figure 3: their current dashed line in Figure 3 states that “Greater distance from the origin beyond this dotted line denotes a higher profile of ecological risk.” Which again, it is based on the assumption that the PI period is what species were led to (or better, evolved/adapted to) expect, and that suddenly in 1850 there was an abrupt shift. In order to make meaningful assumptions over things like that, one needs to also look at longer reconstructions (and not only things like ERA5 as done there, but something like the dataset I provided above).

We thank the reviewer for raising this critical distinction between preindustrial and unforced climates, and agree the unforced preindustrial control is structurally different than the true climate that ecosystems experienced in the past. Thus, we have replaced the previous unforced preindustrial control simulation throughout the manuscript with the CESM2(WACCM6ma) Last Millennium dataset (Otto-Bliesner et al. 2023). The Last Millennium simulation uses boundary conditions from paleoclimate proxies to represent the conditions of the last 1000 years, an approach philosophically similar to conventional simulations of the historical period. This methodology is extensively validated as a Tier 1 experiment (“past1000”) of the Paleoclimate Modelling Intercomparison Project (Jungclaus et al. 2017).

CESM2(WACCM6ma) denotes a “middle atmosphere” configuration of CESM2(WACCM6) with simplified chemistry. Aside from tropospheric chemistry, its representation of global climate is very similar to CESM2(WACCM6) (Davis et al. 2023). The Last Millennium simulation was recently completed and has not yet been documented in an overview publication. However, its immediate predecessor—the CESM1(CAM) Last Millennium Ensemble—successfully reproduced known regional and global features of the paleoclimate record (Otto-Bliesner et al. 2016). Hence, this simulation provides both an ecologically-relevant baseline of preindustrial climate and an appropriate comparison to the CESM2(WACCM6) simulations of future scenarios (SSP2-4.5, ARISE-1.5, ARISE-DelayedStart). We omitted 5 year periods following strong volcanic eruptions (>10 teragrams of stratospheric sulfate injection, Jungclaus et al. 2017). Volcanoes strongly

affect global climate during the Last Millennium, but their perturbations are nonlinear and only last for a few years. Climate velocities are not appropriate for analysis on this timescale due to their definition as a linear regression (e.g., Brito-Morales et al. 2018). In contrast, the SAI simulations we explore feature rapid climate change which is sustained for decades.

Our results with the Last Millennium simulation were qualitatively similar to the original findings with the unforced preindustrial control. In replacing the simulations, we revised a large amount of text throughout Section 1, 2, and the Online Methods as shown in the Tracked Changes file. We summarize the principle results in the response here. Climate speeds were generally small over the average of ten 20-year periods in the Last Millennium (Section 2.1). The distribution of values in ARISE-1.5 and the Last Millennium were not statistically distinguishable (Section 2.2). The Last Millennium baseline for ecological risk in Figure 4 is 10% of global area exposed to climate speeds greater than 10 km/yr, similar to the 11% found in the unforced preindustrial control simulation (Section 2.3).

The unforced preindustrial control is now mentioned in only a few places in our revised manuscript. We include one sentence in Section 2.1 noting the results are qualitatively similar in both the Last Millennium and the unforced simulation. The sentence reads: "A purely unforced simulation with boundary conditions fixed at 1850 (Supplementary Fig. 2) produces qualitatively similar results [to the Last Millennium]." The finding that small climate speeds in the unforced preindustrial control provides confidence in the model's adequacy for our analysis is moved to the Online Methods.

We extensively assessed the literature to find a long-record dataset with full-globe gridded coverage (necessary for the calculation of climate velocities) that provided both a direct comparison to CESM2 simulations and a meaningful baseline for ecology. Unfortunately, the Pages2k datasets are sparse in space and time; thus, it is not possible to use them to calculate climate velocities. Global paleo-reanalyses and previous paleoclimate simulations (see recent review by Smerdon et al. 2023) provide an appropriate ecological baseline, but would have substantial structural differences from CESM2(WACCM6). While the CESM2(WACCM6ma) Last Millennium simulation covers the past 1000 years rather than 2000 years as the reviewer originally suggested, we feel this still provides a sufficient baseline for both ecology and climate in the presence of realistic forcings and internal climate variability.

Works Cited

- Alabia, I. D., García Molinos, J., Saitoh, S. I., Hirawake, T., Hirata, T., & Mueter, F. J. (2018). Distribution shifts of marine taxa in the Pacific Arctic under contemporary climate changes. *Diversity and Distributions*, 24(11), 1583-1597.
- Brito-Morales, I., Molinos, J. G., Schoeman, D. S., Burrows, M. T., Poloczanska, E. S., Brown, C. J., ... & Richardson, A. J. (2018). Climate velocity can inform conservation in a warming world. *Trends in ecology & evolution*, 33(6), 441-457.
- Burrows, M. T., Schoeman, D. S., Buckley, L. B., Moore, P., Poloczanska, E. S., Brander, K. M., ... & Richardson, A. J. (2011). The pace of shifting climate in marine and terrestrial ecosystems. *Science*, 334(6056), 652-655.
- Burrows, M. T., Schoeman, D. S., Richardson, A. J., Molinos, J. G., Hoffmann, A., Buckley, L. B., ... & Poloczanska, E. S. (2014). Geographical limits to species-range shifts are suggested by climate velocity. *Nature*, 507(7493), 492-495.
- Davis, N. A., Visioni, D., Garcia, R. R., Kinnison, D. E., Marsh, D. R., Mills, M., ... & Vitt, F. (2023). Climate, variability, and climate sensitivity of "Middle atmosphere" chemistry configurations of the community Earth system model version 2, whole atmosphere community climate model version 6 (CESM2 (WACCM6)). *Journal of Advances in Modeling Earth Systems*, 15(9), e2022MS003579.
- Ettinger, A. K., & HilleRisLambers, J. (2013). Climate isn't everything: competitive interactions and variation by life stage will also affect range shifts in a warming world. *American Journal of Botany*, 100(7), 1344-1355.
- Jungclaus, J. H., Bard, E., Baroni, M., Braconnot, P., Cao, J., Chini, L. P., ... & Zorita, E. (2017). The PMIP4 contribution to CMIP6–Part 3: The last millennium, scientific objective, and experimental design for the PMIP4 past1000 simulations. *Geoscientific Model Development*, 10(11), 4005-4033.
- MacMartin, D. G., Visioni, D., Kravitz, B., Richter, J. H., Felgenhauer, T., Lee, W. R., ... & Sugiyama, M. (2022). Scenarios for modeling solar radiation modification. *Proceedings of the National Academy of Sciences*, 119(33), e2202230119.
- Otto-Bliesner, B. et al. CESM2-WACCM6ma Last Millennium (2023). Dataset. doi.org/10.26024/5dgt-qr16
- Otto-Bliesner, B. L., Brady, E. C., Fasullo, J., Jahn, A., Landrum, L., Stevenson, S., ... & Strand, G. (2016). Climate variability and change since 850 CE: An ensemble approach with the Community Earth System Model. *Bulletin of the American Meteorological Society*, 97(5), 735-754.
- Richter, J. H., Visioni, D., MacMartin, D. G., Bailey, D. A., Rosenbloom, N., Dobbins, B., ... & Lamarque, J. F. (2022). Assessing Responses and Impacts of Solar climate intervention on the Earth system with stratospheric aerosol injection (ARISE-SAI): Protocol and initial results from the first simulations. *Geoscientific Model Development*, 15(22), 8221-8243.
- Smerdon, J. E., Cook, E. R., & Steiger, N. J. (2023). The Historical Development of Large-Scale Paleoclimate Field Reconstructions Over the Common Era. *Reviews of Geophysics*, 61(4), e2022RG000782.
- Xing, C., Stevenson, S., Fasullo, J., Harrison, C. S., Chen, C. C., Wan, J., ... & Pflieger, C. (2023). Why does Marine Cloud Brightening shut down El Niño–Southern Oscillation?. *AGU23*.

REVIEWERS' COMMENTS

Reviewer #1 (Remarks to the Author):

The authors have addressed all of my comments satisfactorily and I am pleased to recommend this outstanding and important paper. I thought the figures with the actual vectors were very informative, and would like to see at least the ones in the response to reviews incorporated into the supplementary information if possible. The comments by the second reviewer (which I was privileged to see) were interesting; I have never seen an ecological climate change paper that set the baseline at the past millenium--interesting and makes some sense, perhaps, but not the norm.

Reviewer #2 (Remarks to the Author):

The authors have done a great job with the revision. I am grateful they took my suggestion and included the Last Millennium results, and think this greatly strengthened the manuscript. I also like how the figures have been revised.

As I already noted before, the manuscript is excellently written and this hasn't changed in the revision, so I have no further comments. Congratulations on a really, really good and useful paper.

Best,
Daniele Vioni

Response to reviewers for Hueholt et al. “Speed of environmental change frames relative ecological risk in climate change and climate intervention scenarios” [Paper NCOMMS-23-51671-T]

(Previously titled: “Climate speeds help frame relative ecological risk in future climate change and stratospheric aerosol injection scenarios”)

General response

We thank both reviewers for their thoughtful comments during the review process and kind remarks on the revised manuscript. We briefly address their final points below.

Point-by-point response to Reviewer 1

The authors have addressed all of my comments satisfactorily and I am pleased to recommend this outstanding and important paper. I thought the figures with the actual vectors were very informative, and would like to see at least the ones in the response to reviews incorporated into the supplementary information if possible. The comments by the second reviewer (which I was privileged to see) were interesting; I have never seen an ecological climate change paper that set the baseline at the past millenium—interesting and makes some sense, perhaps, but not the norm.

We now include the vector maps as Supplementary Fig. 11. We revised Section 4.2 to introduce this figure: “The climate velocity is a vector quantity, with both a magnitude and a direction. The scalar magnitude alone (climate speed) can be used separately from the vector quantity to quantify the high-level degree of disturbance to ecosystems [18, 34, 79, 80]. This degree of disturbance is the quantity of interest for our research questions, and we use the climate speed exclusively in our analysis. We provide climate velocity vector maps for additional context (Supplementary Fig. 11), however, we caution readers that local analysis of these vectors would require a much finer-resolution dataset to better capture spatial gradients [15, 16, 40, 80].”

Point-by-point response to Reviewer 2

The authors have done a great job with the revision. I am grateful they took my suggestion and included the Last Millennium results, and think this greatly strengthened the manuscript. I also like how the figures have been revised.

As I already noted before, the manuscript is excellently written and this hasn't changed in the revision, so I have no further comments. Congratulations on a really, really good and useful paper.

Best,
Daniele Visioni

We are happy to see that the reviewer likes our redesigned figures and approves of the inclusion of the Last Millennium—we agree this strengthened our analysis.